# Non-perturbative terahertz high-harmonic generation in the three-dimensional Dirac semimetal Cd₃As₂

Sergey Kovalev [1,7], Renato M. A. Dantas [2,7], Semyon Germanskiy [3], Jan-Christoph Deinert [1], Bertram Green[1], Igor Ilyakov[1], Nilesh Awari [1], Min Chen[1], Mohammed Bawatna[1], Jiwei Ling[4,5], Faxian Xiu [4,5], Paul H. M. van Loosdrecht[3], Piotr Surówka[2], Takashi Oka[2,6✉] & Zhe Wang [1,3✉]

Harmonic generation is a general characteristic of driven nonlinear systems, and serves as an efficient tool for investigating the fundamental principles that govern the ultrafast nonlinear dynamics. Here, we report on terahertz-field driven high-harmonic generation in the three-dimensional Dirac semimetal Cd₃As₂ at room temperature. Excited by linearly-polarized multi-cycle terahertz pulses, the third-, fifth-, and seventh-order harmonic generation is very efficient and detected via time-resolved spectroscopic techniques. The observed harmonic radiation is further studied as a function of pump-pulse fluence. Their fluence dependence is found to deviate evidently from the expected power-law dependence in the perturbative regime. The observed highly non-perturbative behavior is reproduced based on our analysis of the intraband kinetics of the terahertz-field driven nonequilibrium state using the Boltzmann transport theory. Our results indicate that the driven nonlinear kinetics of the Dirac electrons plays the central role for the observed highly nonlinear response.

[1] Helmholtz-Zentrum Dresden-Rossendorf, Dresden, Germany. [2] Max Planck Institute for the Physics of Complex Systems, Dresden, Germany. [3] Institute of Physics II, University of Cologne, Cologne, Germany. [4] State Key Laboratory of Surface Physics and Department of Physics, Fudan University, Shanghai, China. [5] Collaborative Innovation Center of Advanced Microstructures, Nanjing, China. [6] Max Planck Institute for Chemical Physics of Solids, Dresden, Germany. [7] These authors contributed equally: Sergey Kovalev, Renato M. A. Dantas. ✉email: oka@pks.mpg.de; zhewang@ph2.uni-koeln.de

In atomic gases[1], high-harmonic radiation is produced via a three-step process of ionization, acceleration, and recollision by a strong-field infrared laser. This mechanism has been intensively investigated in the extreme ultraviolet and soft X-ray regions[2,3], forming the basis of attosecond research[1]. In solid-state materials, which are characterized by crystalline symmetry and strong interactions, yielding of harmonics has just recently been reported[4-20]. The observed high-harmonic generation was interpreted with fundamentally different mechanisms, such as interband tunneling combined with dynamical Bloch oscillations[4,5,7-12,21,22], intraband thermodynamics[16] and nonlinear dynamics[23], and many-body electronic interactions[6,15,17-19,24]. Here, in a distinctly different context of a three-dimensional Dirac semimetal, we report on experimental observation of high-harmonic generation up to the seventh order driven by strong-field terahertz pulses. The observed non-perturbative high-harmonic generation is interpreted as a generic feature of terahertz-field-driven nonlinear intraband kinetics of Dirac fermions. We anticipate that our results will trigger great interest in detection, manipulation, and coherent control of the nonlinear response in the vast family of three-dimensional Dirac and Weyl materials.

High-harmonic generation (HHG) in two-dimensional Dirac semimetals (single-layer graphene[14,16,17] and 45-layer graphene[7]) has been reported very recently for pump pulses both in the terahertz ($10^{12}$ Hz, 1 THz ~4 eV)[7,16] and mid-infrared or near-infrared (0.2–0.8 eV) ranges[14,17]. Although previous theoretical investigations pointed out that the peculiar linear energy-momentum dispersion relation (Dirac cone) should be essential for HHG in graphene (see e.g., ref. [25-27]), the strong dependence on pump laser frequencies observed in the experiments favors different mechanisms. For the mid-infrared or near-infrared HHG, the interband transitions (combined with Bloch oscillations) play the crucial role, while the linear dispersion relation is not a prerequisite[14]. A similar mechanism involving interband transitions can also be applied to terahertz (THz) HHG in lightly-doped multi-layer graphene, whereas the exact shape of the carrier distribution was found to play only a minor role[7]. In contrast, for heavily electron-doped graphene, intraband processes become important and HHG was ascribed to THz-field heated hot-electrons while assuming the electron subsystem thermalized quasi-instantaneously[16].

One may expect to observe THz HHG universally in the Dirac materials also of higher dimension, e.g., three-dimensional (3D) Dirac or Weyl semimetals. However, THz HHG so far has not been reported for this class of materials, and the mechanism for observing THz HHG in a 3D Dirac material remains elusive. Here, we report on time-resolved detection of non-perturbative THz HHG in the 3D Dirac semimetal Cd₃As₂, and a real-time theoretical analysis of the THz-field driven kinetics of the Dirac fermions that is directly linked to the linear dispersion relation. Our results show that the THz-field driven nonlinear kinetics of the Dirac electrons is the mechanism responsible for the efficient generation of high-harmonic radiation, as well as for its non-perturbative fluence dependence in Cd₃As₂.

## Results

**Third harmonic generation.** As being both theoretically predicted and experimentally confirmed[28-33], Cd₃As₂ is a well-established room-temperature 3D Dirac semimetal with Fermi velocity about $10^5$ to $10^6$ m/s. Very compelling topological properties such as topological surface states and 3D quantum Hall effects have been realized in this system[34-38]. In high-quality Cd₃As₂ thin films prepared by molecular beam epitaxy[39], we observe HHG unprecedentedly up to the seventh order in the

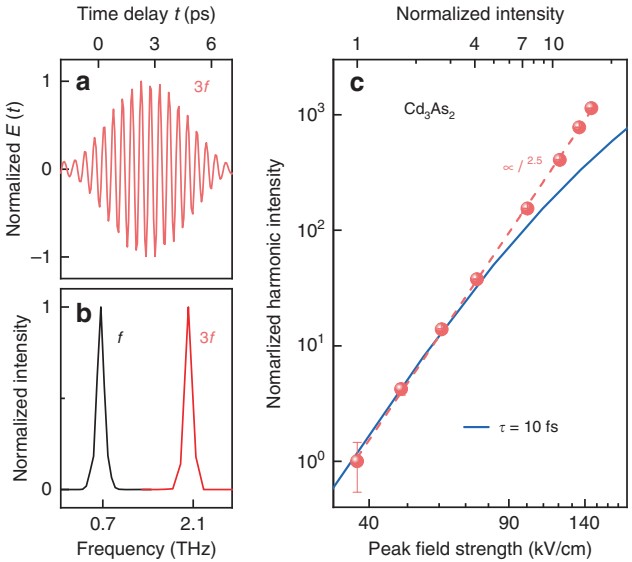

**Fig. 1 Third harmonic generation in Cd₃As₂. a** Time-resolved third-harmonic radiation characterized by its time-dependent electric field $E(t)$ recorded at room temperature. **b** Normalized power spectra of the harmonic radiation $3f = 2.01$ THz, and the excitation pulse $f = 0.67$ THz. **c** Dependence of the third-harmonic radiation intensity on the pump intensity (symbols) follows $I_{3f} \propto I_f^{2.5}$ (dashed line). Fit of the theoretical results is shown for the relaxation time $\tau = 10$ fs (solid line). The error bars indicate the noise level at the corresponding data point.

non-perturbative regime. THz harmonic radiation was recorded with femtosecond resolution at room temperature. Figure 1a displays the detected electric field as a function of time delay for the third harmonic radiation, induced by a multi-cycle pump pulse (Fig. 2a) with a peak field of 144 kV/cm characterized by its central frequency of $f = 0.67$ THz (Fig. 1b). The power spectrum of the harmonic radiation is obtained by Fourier transformation of the time-domain signals, which exhibits a sharp peak at $3f = 2.01$ THz (Fig. 1b). The intensity of the harmonic radiation is nearly independent on the polarization of the pump pulse within the sample surface (see Supplementary Fig. 2). To further characterize the third harmonic generation, we measured the time-resolved signals for different pump-pulse intensities. As summarized in Fig. 1c, the fluence dependence of the third harmonic radiations remarkably does not follow the cubic law, but exhibits a power-law dependence as $I_{3f} \propto I_f^{2.5}$ on the pump-pulse intensity $I_f$, which reveals a non-perturbative nonlinear response.

**THz driven nonlinear kinetics.** To understand the non-perturbative harmonic generation, we performed real-time theoretical analysis of the THz driven kinetics of the 3D Dirac electrons. For the electron-doped system, interband electronic excitations are Pauli-blocked for one-photon transitions in the THz frequency range, thus we focus on the intraband kinetics of the nonequilibrium state by adopting a statistical approach of the Boltzmann transport theory. The initial state of thermodynamic equilibrium is defined by the room-temperature Fermi-Dirac distribution $f_0[\epsilon(\mathbf{p})] = \left[1 + e^{\frac{\epsilon(\mathbf{p})-\epsilon_F}{k_B T}}\right]^{-1}$ for the 3D Dirac electrons obeying the linear dispersion relation $\epsilon(\mathbf{p}) = v_F|\mathbf{p}|$, with $\mathbf{p}$ and $\mathbf{v}_F$ denoting momentum and Fermi velocity, respectively, $\epsilon_F$ for Fermi energy, $k_B$ the Boltzmann constant, and $T$ for temperature. In presence of the THz pulse, the driven transient state is characterized by the distribution function $f(t, \mathbf{p})$, the time-dependent

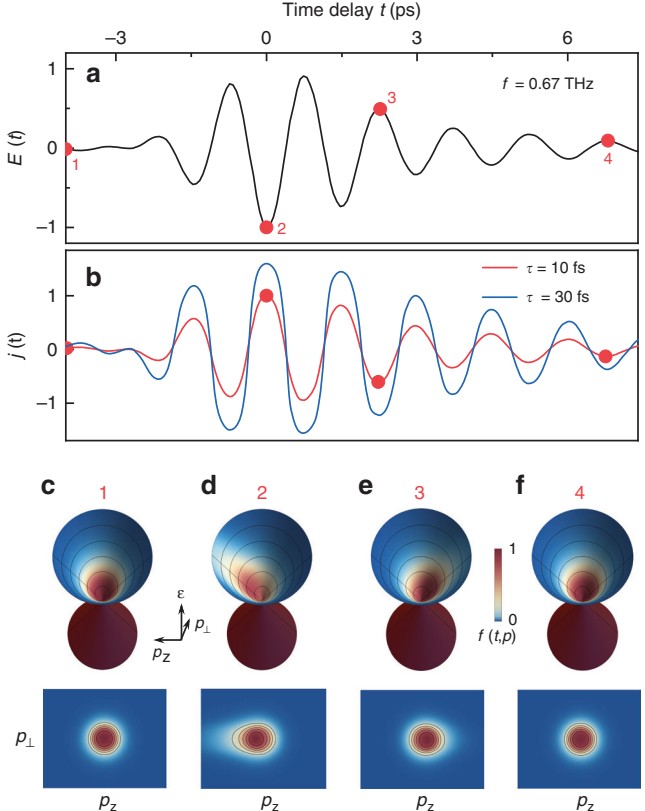

**Fig. 2 THz-driven nonlinear kinetics and time-resolved distribution function. a** Multicycle pump pulse of $f = 0.67$ THz characterized in air by its time-dependent electric field $E(t)$. **b** The derived current density $j(t)$ by solving the Boltzmann equation for $\tau = 10$ and 30 fs, respectively, for the pump pulse with peak field strength of 110 kV/cm. **c–f** 3D and 2D illustration of the distribution function $f(t,p)$ in the upper band of the Dirac cone, corresponding to $\tau = 10$ fs for various time-delays as marked by the points 1–4 in **a**, **b** respectively. $\varepsilon$ denotes energy. $p_z$ denotes momentum component along the linearly-polarized pump-pulse electric field. $p_\perp$ denotes momentum in the perpendicular direction. See Supplementary Fig. 5 and Supplementary Movies 1 and 2 for more comparisons between $\tau = 10$ and 30 fs.

evolution of which is governed by the Boltzmann equation[40,41]

$$\left(\frac{\partial}{\partial t} + \frac{1}{\tau}\right)f(t,\mathbf{p}) - e\mathbf{E}(t) \cdot \nabla_\mathbf{p} f(t,\mathbf{p}) = \frac{f_0(\mathbf{p})}{\tau}, \qquad (1)$$

where the linear dispersion relation has been implemented, $e$ and $\mathbf{E}(t)$ denote the electron charge and the THz electric field, respectively, and $\tau$ is the characteristic relaxation time for intraband processes, which is a phenomenological parameter (see "Methods"). In particular, we do not presume that the electron subsystem thermalizes quasi-instantaneously or a Fermi-Dirac distribution should be obeyed by the transient states. In contrast, by solving the Boltzmann equation, we obtain the real-time distribution of the transient state. By comparing it with the equilibrium-state Fermi-Dirac distribution, we can claim whether the corresponding transient state is nearly thermalized or far from thermodynamic equilibrium. Furthermore, we can derive the time-dependent current density, hence the THz field-induced harmonic radiation, the fluence dependence of which can be compared to the experimental observations.

For the experimentally implemented THz pump pulses (see Fig. 2a for the waveform) with a typical electric-field peak

strength of 110 kV/cm, the obtained current density (Fig. 2b) and transient-state distribution functions are illustrated in Fig. 2c–f, corresponding to the representative time delays (red symbols) marked in Fig. 2a, b, for the experimental values of Fermi energy $\varepsilon_F = 118$ meV and Fermi velocity $v_F = 7.8 \times 10^5$ m/s as estimated from Shubnikov-de Haas oscillations[39], and the relaxation time $\tau = 10$ fs. The electric field of the linearly-polarized pump pulse is set along the $p_z$ direction.

The microscopic origin of HHG resides in the nonlinear kinetics of the electron distribution (see Fig. 2d–f and Supplementary Fig. 5) combined with the linear energy–momentum dispersion relation. Before the pump pulse arrives, the electrons in the upper band are in thermodynamic equilibrium, and fill the Dirac-cone up to around the Fermi energy according to the Fermi-Dirac distribution (Fig. 2c). When the pump pulse is present, the electrons are not only accelerated by the THz electric fields, but at the same time also scattered. Although the latter process is dissipative, the former one can very efficiently accumulate energy into the electron subsystem, leading to a stretched and shifted distribution along the field. In particular, at the peak field (symbol point 2 marked in Fig. 2a), the distribution is most strongly stretched and shifted in the field direction (Fig. 2d) resulting in the maximum current density and a peculiar flat-peak-like feature (Fig. 2b), thereby leading to very efficient HHG. In clear contrast to the Fermi-Dirac distribution of a thermodynamic equilibrium state that is spherically symmetric for the 3D Dirac electrons (manifested as circularly symmetric in the 2D plots); the obtained strongly stretched and highly asymmetric distribution due to the presence of the strong THz field evidently shows that the electron subsystem is far from thermodynamic equilibrium. As shown in Fig. 2f, the electron distribution becomes nearly symmetric in low THz fields, indicating that a quasi-thermalized situation is reconciled in the low-field limit.

For various pump-pulse peak field strength, the intensity of the third-harmonic radiation is shown in Fig. 1c for relaxation time $\tau = 10$ fs. The peak field strength in the sample is estimated as the average value over the film thickness. The theoretical results reproduce excellently the observed non-perturbative fluence dependence of the third-harmonic generation up to about 80 kV/cm of the peak field strength, though a deviation from the experimental data occurs at higher fluences. This deviation could be due to enhanced probability of interband multiphoton tunneling in the high electric-field limit, which is not included in our semi-classical analysis. Nevertheless, we found that the non-perturbative dependence on pump-pulse fluence is a generic feature of the THz driven nonequilibrium states in the Dirac semimetals. Furthermore, we found that efficiency and fluence-dependence of the THz HHG is very sensitive to the scattering rate $1/\tau$. By decreasing the scattering rate (or suppressing the dissipative processes), the transient distribution function is further stretched for the same electric-field strength, resulting in greater current density (c.f. $\tau = 30$ fs in Fig. 2b) and enhanced THz HHG (see Supplementary Fig. 3, Supplementary Fig. 4, Supplementary Fig. 5, and Supplementary Movie 1 Supplementary Movie 2 for the real-time evolution of the distribution driven by the THz pulse in Fig. 2a). Our theoretical calculations further reveal that for a fixed scattering rate the harmonic generation is enhanced at a higher Fermi energy (see Supplementary Fig. 6), which is compelling for further experimental studies.

**Higher-order harmonic generation.** In order to detect higher-order harmonic radiation, we utilized lower-frequency and strong-field THz pump pulses (see "Methods")[42]. Figure 3a shows the observed harmonic radiation up to the seventh order for the pump-pulse frequency of 0.3 THz (see Fig. 3b for the waveform).

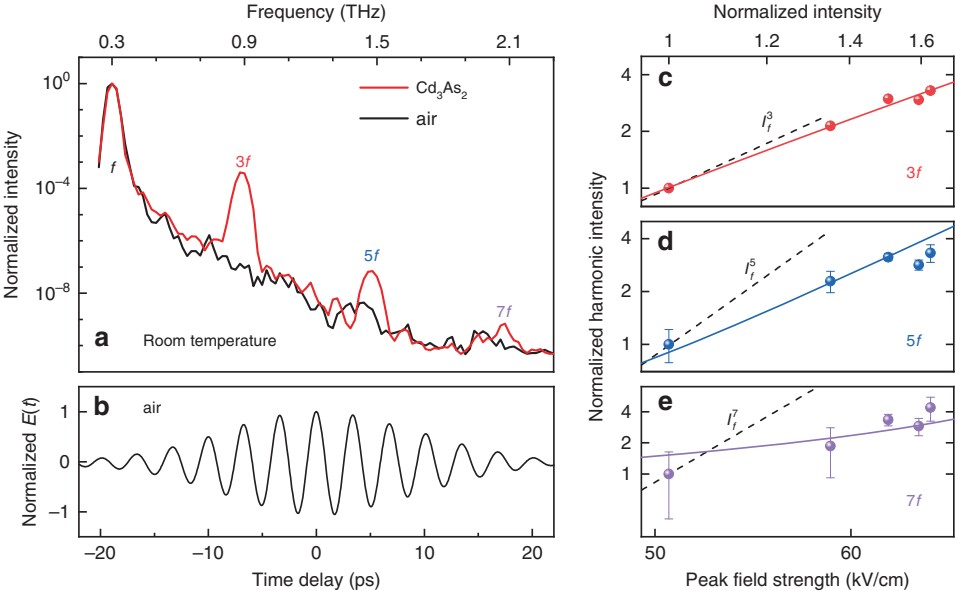

**Fig. 3 High-harmonic generations in Cd$_3$As$_2$. a** Room-temperature spectrum of high-harmonic generations in Cd$_3$As$_2$ for **b** multicycle pump pulse of $f = 0.3$ THz, compared with air as a reference. Pump-intensity dependence of the **c** third-harmonic, **d** fifth-harmonic and **e** seventh-harmonic generation (symbols) follows the power laws of $I_f^{2.6\pm0.1}$, $I_f^{2.8\pm0.1}$, and $I_f^{2.8\pm0.7}$, respectively. The dashed lines indicate the corresponding perturbative power laws, i.e., $\propto I_f^3$, $I_f^5$, and $I_f^7$. In **c**–**e**, the solid lines show the fitted theoretical results corresponding to the relaxation time $\tau = 10$ fs. The error bars indicate the noise level at the corresponding data point.

Only the odd-order harmonics are observed, providing the spectroscopic evidence for the existence of inversion symmetry in the crystalline structure of Cd$_3$As$_2$ (see ref. [33]). Our experimental results not only set the record for THz HHG in the 3D Dirac materials, but also present the striking observation of the non-perturbative fluence dependence for all the observed harmonic orders, as presented in Fig. 3c–e.

For the third harmonic radiation, the fluence dependence is also slightly below the cubic power-law dependence, similar to the behavior for the 0.7 THz pump pulse. Moreover, for the higher-order harmonics, the deviation from the corresponding perturbative power-law dependence is further increased. These features are perfectly captured by our quantitative theoretical analysis. By implementing the experimental pump pulse (see Fig. 3b) in our calculations, the time-resolved harmonic signals are derived as a function of pump-pulse fluence. The best fitting for all the experimentally observed HHG is achieved at $\tau = 10$ fs (see Fig. 3c–e). The obtained value of $\tau = 10$ fs is comparable to that in graphene as directly obtained via time-resolved and angle-resolved photoemission spectroscopic measurements[43]. While such measurements have not been reported in Cd$_3$As$_2$, an estimate based on the Shubnikov-de Haas measurements provides a $\tau$ value of the same order[39]. These results strongly indicate that the THz field-driven nonlinear kinetics of the Dirac electrons is the mechanism responsible for the observed non-perturbative nonlinear response in Cd$_3$As$_2$. Although for the seventh harmonic the experimental uncertainty is enhanced at the lowest fluence, the fluence dependence far away from the perturbative one is a clear and consistent experimental and theoretical observation. The non-perturbative response could be qualitatively understood in a way that the effective nonlinear susceptibilities are also function of the THz field due to the higher-order nonlinear response. We note that the observed non-perturbative response suggests that the experimental setting is close to but still below the so-called high-harmonic plateau regime, in which the HHG intensity remains almost constant for the high orders and drops abruptly at a cutoff frequency as found in gases as well as in solids[1,20].

## Discussion

The established mechanism of THz HHG here based on the driven nonlinear kinetics of Dirac electrons is different from those mechanisms proposed for HHG in graphene[7,14,16,17], in which either the interband transitions were found playing the dominant role or the intraband electron subsystem is assumed to thermalize quasi-instantaneously. In contrast, in the context of the 3D Dirac system, we found that, firstly, in the presence of strong THz fields, the entire intraband distribution is strongly stretched and highly asymmetric, denying a description using the Fermi-Dirac distribution of thermodynamic equilibrium states that is symmetric along the Dirac cone. Secondly, for the intraband kinetics, the linear energy-momentum dispersion is crucial for the THz HHG, whereas for a parabolic dispersion in the single-particle picture, the induced radiation field $E_{out} \propto \frac{dj}{dt} \propto \frac{dv}{dt} \propto E_{in}$, should follow the pump field $E_{in}$, hardly yielding harmonics. Thirdly, the exact shape of the electron distribution and its real-time evolution, as obtained from the Boltzmann transport theory, is directly responsible for the THz HHG. A higher efficiency is revealed for the cases of a more strongly stretched and highly asymmetric distribution, due to stronger THz electric field and/or reduced scattering rate.

In conclusion, we have observed THz driven HHG up to the seventh order unprecedentedly in the 3D Dirac semimetal Cd$_3$As$_2$. The fluence dependence of all the observed HHG was found well beyond the perturbative regime. By performing real-time quantitative analysis of the THz field-driven intraband kinetics of the Dirac electrons using the Boltzmann transport theory, we have established the nonlinear intraband kinetics as the mechanism for the observed THz HHG in Cd$_3$As$_2$. The mechanism found here for THz HHG is expected to be universal in the vast family of 3D Dirac and Weyl materials[44], which provides strategies for pursuing high efficiency of THz HHG, and establishes HHG as a sensitive tool for exploring the interplay of various degrees of freedom. Towards the high electric-field regime, an experimental realization of THz HHG plateau in the Dirac materials and a full quantum-mechanical dynamic analysis are still outstanding from both the fundamental and the application points of view. Recently, non-perturbative THz third-harmonic generation in Cd$_3$As$_2$ was also reported in Ref. [45].

## Methods

**Terahertz spectroscopy**. We performed terahertz THz HHG experiments with THz sources based on a femtosecond laser system and on a linear electron accelerator. For the former, broadband THz radiation was generated through tilted pulse front scheme utilizing lithium niobate crystal[46–48]. With initial laser pulse energy around 1.5 mJ at 800 nm central wavelength and 100 fs pulse duration broadband THz radiation with up to 3 μJ pulse energy was generated. At the linear accelerator in Helmholtz Zentrum Dresden-Rossendorf, multi-cycle superradiant THz pulses were generated in an undulator from ultra-short relativistic electron bunches[42]. The generated THz radiation is carrier envelope phase stable, linear polarized with tunable emitted radiation frequency. The accelerator was operated at 100 kHz and was synchronized with an external femtosecond laser system. The latter served as probe in electro-optical sampling. To achieve high level of synchronization, a pulse-resolved detection scheme was employed[49]. To produce narrow-band THz radiation, corresponding bandpass filters were used (see Supplementary Fig. 1 for more information).

**Sample preparation and characterization**. High-quality thin films of $Cd_3As_2$ were grown by PerkinElmer (Waltham, MA) 425B molecular beam epitaxy system[39]. The substrate of fresh-cleaved 2-inch mica (~70 μm in thickness) was annealed at 300 °C for 30 min to remove absorbed molecules. Then 10 nm-thick CdTe was deposited as buffer layer before the $Cd_3As_2$ growth. $Cd_3As_2$ bulk material (99.9999%, American Elements Inc., Los Angeles, CA) was evaporated on to CdTe at 170 °C. The growth was in situ monitored by reflection high-energy electron diffraction (RHEED) system. The sample surface is parallel to the crystallographic (112) plane. Part of the sample was patterned in Hall bar geometry and performed magnetic resistance measurement on physical properties measurement system (PPMS) (Quantum Design Inc.). Fermi energy and Fermi velocity of the 120 nm-thick $Cd_3As_2$ samples was estimated as $E_F = 118$ meV and $v_F = 7.8 \times 10^5$ m/s from the Shubnikov-de Haas oscillations. THz transmission was characterized in the linear response regime by standard electro-optical sampling scheme.

**Kinetic theory**. Our theoretical analysis employed a statistical approach of the semiclassical Boltzmann transport theory with an effective relaxation time[40,41,50–53]. The semiclassical description of particles is captured by a single particle distribution function $f(t, \mathbf{r}, \mathbf{p})$ in phase space. Observables can be calculated as integrals over momentum space. In order to calculate $f(t, \mathbf{r}, \mathbf{p})$ one needs to solve the Boltzmann equation

$$df \equiv \partial_t f + \nabla_\mathbf{r} f \cdot \dot{\mathbf{r}} + \nabla_\mathbf{p} f \cdot \dot{\mathbf{p}} = \mathcal{C}[f].$$

The left hand side of this equation corresponds to the collisionless evolution in phase space. The collision integral can either be calculated perturbatively from scattering amplitudes or chosen phenomenologically. In this work we use phenomenological relaxation time approximation and choose Bhatnagar–Gross–Krook collision operator[40].

The explicit form of the Boltzmann equation follows from the (inverted) equations of motion for the electron's wavepacket[51–53]

$$\dot{\mathbf{r}} = \frac{1}{\hbar D}\left[\nabla_\mathbf{k}\epsilon_\mathbf{k} + e\mathbf{E}\times\mathbf{\Omega} + \frac{e}{\hbar}(\nabla_\mathbf{k}\epsilon_\mathbf{k}\cdot\mathbf{\Omega})\mathbf{B}\right],$$

$$\hbar\dot{\mathbf{k}} = \frac{1}{D}\left[-e\mathbf{E} - \frac{e}{\hbar}\nabla_\mathbf{k}\epsilon_\mathbf{k}\times\mathbf{B} - \frac{e^2}{\hbar}(\mathbf{E}\cdot\mathbf{B})\mathbf{\Omega}\right],$$

with the electromagnetic fields $\mathbf{E}$ and $\mathbf{B}$, the Berry curvature $\mathbf{\Omega}$, the Planck constant $\hbar$, and the elementary charg $e$. $\epsilon_\mathbf{k}$ denotes the dispersion relation and $D = 1 + \frac{e}{\hbar}\mathbf{B}\cdot\mathbf{\Omega}$ is the modified phase space volume element. For the linearly polarized THz pulses, we consider the dominant effects of the electric field while neglecting the magnetic field in our further analysis. Consequently, the (inverted) equations of motion take the following simple form

$$\dot{\mathbf{r}} = \nabla_\mathbf{p}\epsilon_\mathbf{p} + \frac{e}{\hbar}\mathbf{E}\times\mathbf{\Omega}, \quad \dot{\mathbf{p}} = \hbar\dot{\mathbf{k}} = -e\mathbf{E}.$$

Since we are interested in a homogenous solution, only the equation for $\dot{\mathbf{p}}$ is incorporated in the Boltzmann equation. The equation for $\dot{\mathbf{r}}$ is used to define the current density as follows:

$$\mathbf{j}(t) = -e\int \frac{d^3p}{(2\pi\hbar)^3}\dot{\mathbf{r}}f(t, \mathbf{p}).$$

Nevertheless, it can be shown that the second term in this equation (proportional to $\mathbf{E}\times\mathbf{\Omega}$) does not contribute to $\mathbf{j}(t)$ in the case of linearly polarized THz pulse, corresponding to the present experimental setting. Therefore, for the particular experiment being reported now, we can write

$$\dot{\mathbf{r}} = \nabla_\mathbf{p}\epsilon_\mathbf{p}, \quad \dot{\mathbf{p}} = -e\mathbf{E}.$$

For the THz frequencies in our experiments, interband electronic transitions are Pauli-blocked for the electron-doped $Cd_3As_2$ samples. Thus, to study the intraband electron dynamics, it is justified to adopt one relaxation scale. In addition to that the underlying impurities in the system can lead to non-conservation of charge and momentum. As a result, we expect that the collision

integral of the following form

$$\mathcal{C}[f] = \frac{f_0 - f}{\tau},$$

will correctly reproduce the experimental data. In equilibrium the distribution function should depend on collisional invariants

$$f_0(\beta, \mathbf{p}, \epsilon_F) = \left[1 + e^{\beta(v_F|\mathbf{p}| - \epsilon_F)}\right]^{-1},$$

where $\beta \equiv 1/k_B T$ with the Boltzmann constant $k_B$, $\epsilon_F$ denotes the Fermi energy, and the linear dispersion relation $\epsilon_\mathbf{p} = v_F|\mathbf{p}|$ of the Dirac material has been implemented. Finally, considering only homogeneous response, we arrived at the following Boltzmann equation

$$\left(\partial_t + \frac{1}{\tau}\right)f - e\mathbf{E}\cdot\nabla_\mathbf{p}f = \frac{f_0}{\tau},$$

where the external driving force $\mathbf{F} = -e\mathbf{E}$ is implemented for electrons moving in the THz electric field $\mathbf{E}$. In order to solve this equation, we Fourier transform the distribution function $f(t, \mathbf{p}) = \frac{1}{2\pi}\int dz \tilde{f}(t, p_x, p_y, z)\exp(izp_z)$, which gives an ordinary differential equation

$$\left(\partial_t + \frac{1}{\tau}\right)\tilde{f} - izeE\tilde{f} = \frac{\tilde{f_0}}{\tau},$$

where the electric field $\mathbf{E}$ has been set along the $z$ direction.

The ordinary differential equation is solved numerically with the experimental THz fields as an input. Having the distribution function, we calculate its moments to get current density. The expression for current density has the following form

$$\mathbf{j}(t) = -e\int \frac{d^3p}{(2\pi\hbar)^3}v_F\hat{\mathbf{p}}f(t, \mathbf{p}),$$

where $\hat{\mathbf{p}}$ denotes the unit vector along the momentum direction. The relation between the induced current and the external oscillating field serves as the basis for analysis of higher-harmonic generation.

## Data availability

Data supporting the findings of this work are available from the corresponding authors upon reasonable request. Further requests on the raw pre-sorted and pre-binned data should be sent to HZDR via S.K.

## Code availability

Computer code supporting the findings of this works is available from R.M.A.D. upon reasonable request.

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

## Acknowledgements

Parts of this research were carried out at ELBE at the Helmholtz-Zentrum Dresden—Rossendorfe. V., a member of the Helmholtz Association. We would like to thank U. Lehnert, J. Teichert and the rest of the ELBE team for assistance, and M. Gensch for support of the experiments. Z.W. thanks A. Renno for characterizing the samples microscopically, and Enke Liu and C. Reinhoffer for helpful discussions. S.K. and B.G. acknowledge support from the European Cluster of Advanced Laser Light Sources (EUCALL) project, which has received funding from the European Union's Horizon 2020 research and innovation programme under Grant Agreement No 654220. N.A., S.K., and I.I. acknowledge support from the European Union's Horizon 2020 research and innovation program under grant agreement No. 737038 (TRANSPIRE). R.M.A.D. and P.S. were supported in part by the DFG (German Research Foundation) through the Leibniz Program and ct.qmat (EXC 2147, project-id 39085490). F.X. was supported by National Natural Science Foundation of China (Grant No. 11934005, 61322407, 11474058, 61674040 and 11874116), National Key Research and Development Program of China (Grant No. 2017YFA0303302 and 2018YFA0305601). P.v.L. and Z.W. acknowledge support by the DFG via project No. 277146847—Collaborative Research Center 1238: Control and Dynamics of Quantum Materials (Subproject No. B05).

## Author contributions

Z.W. and T.O. conceived the project with P.S. S.K., and Z.W. carried out the THz HHG experiments and analyzed the data with S.G., J.-C.D., B.G., I.I., N.A., M.C., M.B. R.M.A.D., P.S., and T.O. performed the theoretical calculations and analyzed the data. J.L. and F.X. fabricated and characterized the high-quality samples. S.G., P.v.L., and Z.W. characterized linear THz response of the samples. Z.W. wrote the manuscript with contributions from S.K., R.M.A.D., S.G., J.L., P.S., and T.O. All authors commented the manuscript.

## Competing interests

The authors declare no competing interests.
