## [Peer Review File · Nature Communications]

Reviewers' comments:

Reviewer #1 (Remarks to the Author):

The paper describes interesting experiments on HHG in the Dirac semimetal Cd₃As₂. I am in favor of acceptance of the paper provided the following changes have been made prior to publication.

I regard as mandatory:

1. The authors use an extremely simple quasi-classical model basically without any quantum mechanics to describe the obtained results. It is surprising and it is to my opinion an important result of the paper that this model can at least qualitatively reproduce the experimental findings for moderate intensities. The authors should more clearly underline the simplifications of their model and avoid statements as "the analysis is performed without further assumptions".

2. I would like to see a statement on the role of the Fermi energy in the main paper. If the model is correct its position must be crucial for HHG. I can understand that a variation of that parameter might not be accessible in the experiment, but respective simulations should be performed and discussed in the supplementary.

2. Unless I have overlooked it, but the authors should give the power dependence for the different harmonics displayed in Figs 3 c-e (represented by bold lines).

Optionally I recommend the authors to look for a simple explanation for the observed power law dependence of the different harmonics on the driving pump intensity. The paper would gain a lot by such an explanation. However, it might be that there is no simple explanation.

Reviewer #2 (Remarks to the Author):

Reports Kovalev et al.

Non-perturbative terahertz high-harmonic generation in the three-dimensional Dirac semimetal Cd₃As₂

This communication reports the THz high-harmonic generation based on both experiment and theoretical analysis. The experiments are performed using THz pump light (0.7 THz and 0.3 THz) and EO sampling to detect time evolutions of the emitted electric field of the HHG up to 7-th order (for 0.3 THz pump). The authors analyze the nonlinear-transient current reflecting electron distribution function $t(t, p)$ using the Boltzmann transport theory. The authors claim that the THz HHG in Cd₃As₂ is well described by non-equilibrium intra-band charge distribution. They also claim that the observed excitation intensity dependence (which is smaller than those for perturbative power law) is reproducible by the above analysis, indicating the non-perturbative mechanism of the HHG in the Dirac system.

They nicely achieve the observations of the HHG up to 7-th order HHG as real time waveforms using 0.3 THz pump light from a linear electron accelerator which is accurately synchronized with a femtosecond laser. In addition, the real time tracking of the distribution of $f(t, p)$ in the analysis, demonstrating the asymmetric stretching of the electronic distribution toward the direction of the applied THz field (and the resultant flat-peak like feature in the time profile of the current density profile), is very clear to see that the HHG is driven by such non-equilibrium intra-band charge distribution. I agree that, the HHG is successfully in the Dirac electron having a linear dispersion relation).

Another important issues of this manuscript is that they could determine the intra-band relaxation time (in the Boltzmann theory) for reproducing the fullerene dependence. They denote that "Furthermore, we found that efficiency and fluence-dependence of the THz HHG is very sensitive to the scattering rate $1/\tau$ ". That is very important for clarifying the mechanism of HHG in Dirac and Weyl system.

Considering the attracting features of the 3D Dirac and Weyl materials and their possible nonlinear photonics, I agree the THz HHG of this typical Dirac compound can provide large impact both in

the solid-state physics and the photonics communities. Therefore, I recommend publication of this paper. However, while I have no doubts on the very important observations of the HHG and their analysis, I have a few tiny comments and queries (most of them are technical problems).

i) Why do you use asymmetric electric field in the calculation (Fig. 2(a))?. That is different from the THz pulse used in the experiment (Fig. 3(b)).

ii) I think that more detailed discussions can be useful for the intra-band relaxation time. Is it possible to consider the microscopic mechanism of the scattering? The scattering time of 10 fs is likely to be reasonable (as actually shown in graphene (ref 43)). However, I'm not sure that it is really justified also in this compound.

iii) It seems not so clear that the fluence dependence (in the analysis) is sensitive to $1/\tau$ in extended data fig.3, although the authors have said so.

iv) Moreover, I'm also interested in the calculated results for stronger field (such as several MV/cm in the analysis). Is it qualitatively different from those shown here?

Reviewer #1 (Remarks to the Author):

The paper describes interesting experiments on HHG in the Dirac semimetal Cd₃As₂. I am in favor of acceptance of the paper provided the following changes have been made prior to publication.

We would like to thank the reviewer for carefully reviewing our manuscript, and for the positive statement that he/she is in favor of acceptance of the paper. According to the reviewer's suggestion, we have made changes and provide a point-by-point response in the following.

I regard as mandatory:

1. The authors use an extremely simple quasi-classical model basically without any quantum mechanics to describe the obtained results. It is surprising and it is to my opinion an important result of the paper that this model can at least qualitatively reproduce the experimental findings for moderate intensities. The authors should more clearly underline the simplifications of their model and avoid statements as "the analysis is performed without further assumptions".

We thank the reviewer for pointing out that the quantum mechanical effects have not been fully included in the quasi-classical model. We have addressed on Page 3 that "the multiphoton processes in the high electric-field limit is not included in our semi-classical analysis", and also at the end of the manuscript that "a full quantum-mechanical dynamic analysis is still outstanding".

Following the Reviewer's suggestion, we have expanded the description of our theoretical approach including the simplifications in the Methods. We have removed the statement "the analysis is performed without further assumptions" in the main text.

2. I would like to see a statement on the role of the Fermi energy in the main paper. If the model is correct its position must be crucial for HHG. I can understand that a variation of that parameter might not be accessible in the experiment, but respective simulations should be performed and discussed in the supplementary.

Following this suggestion, we have performed additional simulations and illustrated the results in the Supplementary Fig. 6. In the main text, we have added on Page 3, at the end of the 4th paragraph, that "Our theoretical calculations further reveal that for a fixed scattering rate the harmonic generation is enhanced at a higher Fermi energy (see Supplementary Fig. 6), which is compelling for further experimental studies."

3. Unless I have overlooked it, but the authors should give the power dependence for the different harmonics displayed in Figs 3 c-e (represented by bold lines).

Following this suggestion, we have added in the caption of Fig. 3 the power dependence for the different harmonics, which reads "Pump-intensity dependence of the **c**, third-, **d**, fifth-, and **e**, seventh-harmonic generation (symbols) follows the power laws of $I_f^{2.6\pm 0.1}$, $I_f^{2.8\pm 0.1}$, and $I_f^{2.8\pm 0.7}$, respectively."

Optionally I recommend the authors to look for a simple explanation for the observed power law dependence of the different harmonics on the driving pump intensity. The paper would gain a lot by such an explanation. However, it might be that there is no simple explanation.

So far we have not been able to provide analytical results to explain the observation. Qualitatively, we could understand the non-perturbative fluence dependence as a consequence that the effective nonlinear susceptibility is also dependent on the THz electric field. In general, we could write the emitted THz field as a power expansion of all the odd harmonics, i.e.

$$E_{out} \sim \chi^{(1)} E_{in} + \chi^{(3)} E_{in}^3 + \chi^{(5)} E_{in}^5 + \chi^{(7)} E_{in}^7 + \dots$$

with the susceptibility $\chi^{(n)}$, $n = 1, 3, 5, \dots$

For the THz electric field $E_{in} = E_0 \cos \omega t$, we can rewrite

$$E_{out}(t) \sim \chi^{(1)} E_0 \cos \omega t + \chi^{(3)} E_0^3 \cos^3 \omega t + \chi^{(5)} E_0^5 \cos^5 \omega t + \chi^{(7)} E_0^7 \cos^7 \omega t + \dots$$

Since the expansion of the $\cos^n \omega t$ term does not only contain the frequency component of $n\omega$ but also all the lower frequency components, i.e. $(n-2)\omega$, $(n-4)\omega$, ..., ω , the overall 3ω component obtained via Fourier transformation of $E_{out}(t)$ has the general form of

$$E_{3\omega} \sim \chi^{(3)} E_0^3 + a_{53} \chi^{(5)} E_0^5 + a_{73} \chi^{(7)} E_0^7 + \dots$$

where a_{53} , a_{73} etc are coefficients. Then, the effective third-order nonlinear susceptibility

$$\chi_{eff}^{(3)}(E_0) \sim \chi_0^{(3)} + a_{53} \chi^{(5)} E_0^2 + a_{73} \chi^{(7)} E_0^4 + \dots$$

is also a function of the THz electric field, reflecting the non-perturbative nature of the nonlinear response. For the linear dispersion relation, the higher-order susceptibilities $\chi^{(n)}$'s are enhanced thereby enhancing the non-perturbative behavior. In contrast, for the quadratic dispersion relation, the $\chi^{(n)}$'s can be negligible so the response of the conventional systems stay in the perturbative regime. In the revised manuscript, we have added one sentence on Page 4 to express this idea: "The non-perturbative response could be qualitatively understood in a way that the effective nonlinear susceptibilities are also function of the THz field due to the higher-order nonlinear response".

Reviewer #2 (Remarks to the Author):

This communication reports the THz high-harmonic generation based on both experiment and theoretical analysis. The experiments are performed using THz pump light (0.7 THz and 0.3 THz) and EO sampling to detect time evolutions of the emitted electric field of the HHG up to 7-th order (for 0.3 THz pump). The authors analyze the nonlinear-transient current reflecting electron distribution function $f(t, p)$ using the Boltzmann transport theory. The authors claim that the THz HHG in Ca_3As_2 is well described by non-equilibrium intra-band charge distribution. They also claim that the observed excitation intensity dependence (which is smaller than those for perturbative power low) is reproducible by the above analysis, indicating the non-perturbative mechanism of the HHG in the Dirac system.

They nicely achieve the observations of the HHG up to 7-th order HHG as real time waveforms using 0.3 THz pump light from a linear electron accelerator which is accurately synchronized with a femtosecond laser. In addition, the real time tracking of the distribution of $f(t, p)$ in the analysis, demonstrating the asymmetric stretching of the electronic distribution toward the direction of the applied THz field (and the resultant flat-peak like feature in the time profile of the current density profile), is very clear to see that the HHG is driven by such non-equilibrium intra-band charge distribution. I agree that, the HHG is successfully in the Dirac electron having a linear dispersion relation).

Another important issues of this manuscript is that they could determine the intra-band relaxation time (in the Boltzmann theory) for reproducing the fullerene dependence. They denote that "Furthermore, we found that efficiency and fluence-dependence of the THz HHG is very sensitive to the scattering rate $1/\tau$ ". That is very important for clarifying the mechanism of HHG in Dirac and Weyl system.

Considering the attracting features of the 3D Dirac and Weyl materials and their possible nonlinear photonics, I agree the THz HHG of this typical Dirac compound can provide large impact both in the solid-state physics and the photonics communities. Therefore, I recommend publication of this paper.

However, while I have no doubts on the very important observations of the HHG and their analysis, I have a few tiny comments and queries (most of them are technical problems).

We would like to thank the reviewer for the very positive report and the recommendation for publication of the paper. We are very happy to read the statement that *the THz HHG of this typical Dirac compound can provide large impact both in the solid-state physics and the photonics communities*. In the following, we provide a point-by-point response to the comments and queries.

i) Why do you use asymmetric electric field in the calculation (Fig. 2(a))? That is different from the THz pulse used in the experiment (Fig. 3(b)).

The pulses shown in Fig.2(a) and Fig.3(b) are both experimental data, corresponding to 0.67 and 0.3 THz, respectively, as given in the figure captions. It is just a matter of experimental fact that their waveforms in time domain are asymmetric or symmetric. Both of them have been used in our calculations to understand the respective experimental results.

ii) I think that more detailed discussions can be useful for the intra-band relaxation time. Is it possible to consider the microscopic mechanism of the scattering? The scattering time of 10 fs is likely to be reasonable (as actually shown in graphene (ref 43)). However, I'm not sure that it is really justified also in this compound.

The intra-band relaxation time is a phenomenological parameter, which is not given *a priori* but obtained by comparing to the experimental results. A direct measurement of the scattering time could be made using time- and angle-resolved photoemission spectroscopy, as done in Ref. 43 for graphene. However, such measurement has not been reported in Cd₃As₂. An estimate of the scattering time can be given based on the the Shubnikov-de Haas measurements (see Ref. 39), which is comparable to the value we obtained here. We have added these discussions in the revised manuscript.

iii) It seems not so clear that the fluence dependence (in the analysis) is sensitive to $1/\tau$ in extended data fig.3, although the authors have said so.

When we compare the effects of varying tau, we should fix the other parameters. As shown in the supplementary fig.3, in the strong-field regime (e.g. 80 kV/cm), the slope (or the fluence dependence) of the curve for $\tau = 10$ fs is clearly smaller than that for $\tau = 30$ fs. The low-field regime is not relevant here, because the HHG there is inefficient and negligible.

iv) Moreover, I'm also interested in the calculated results for stronger field (such as several MV/cm in the analysis). Is it qualitatively different from those shown here?

According to our quasi-classical analysis, the HHG increases with increasing field and eventually saturates in the strong-field limit (see Supplementary Fig.3 curve of $\tau = 30$ fs). However, as we noted in the manuscript (see discussion on Fig.1c), quantum effects (such as inter-band transition due to multi-photon processes) could be dominant in the strong-field limit, which is not included in our quasi-classical analysis. As pointed out in the last paragraph, a full quantum mechanical analysis is outstanding for a further study.

REVIEWERS' COMMENTS

Reviewer #1 (Remarks to the Author):

The authors have followed all the advices of the referees and have successfully revised the manuscript. I have no further objections against a rapid publication.

Reviewer #2 (Remarks to the Author):

Reports Kovalev et al.

Non-perturbative terahertz high-harmonic generation in the three[^]dimensional Dirac semimetal Cd₃As₂

2nd report:

I read the response of the authors and the revised manuscript and find that it improved and clarifies most of the raised questions for both reviews. I confirm that the points I raised in the previous round have been satisfactorily addressed. I am enthusiastic by the novelty of this paper and I strongly recommend its publication in Nature Communications.